# Spatial Justice and Residents' Policy Acceptance: Evidence from Construction Land Reduction in Shanghai, China

Keqiang Wang [1,2], Jianglin Lu [1,*], Hongmei Liu [3], Fang Ye [1], Fangbin Dong [4] and Xiaodan Zhu [4]

1. School of Public Economics and Administration, Shanghai University of Finance and Economics, Shanghai 200433, China
2. Technology Innovation Center for Land Spatial Eco-restoration in the Metropolitan Area, MNR, Shanghai 200003, China
3. School of Finance and Business, Shanghai Normal University, Shanghai 200233, China
4. Centre for Shanghai Municipal Construction Land and Land Consolidation, Shanghai 200003, China
* Correspondence: lujianglin@163.sufe.edu.cn

**Abstract:** Nowadays, the contradiction between strict construction land supply restriction and excessive construction land demand is extremely prominent. Construction land reduction (CLR) is a policy innovation for economically developed regions designed to solve the tight constraints of the construction land quota as urban development continues in China, however, it leads to a lack of spatial justice. In this study, we address a gap in land use regulation literature regarding regional economic development in fast-developing nations by presenting a quantitative investigation of spatial justice in Shanghai, China. We theoretically analyze the connotation of spatial justice in CLR and its influence on residents' policy acceptance of CLR. Based on theoretical analysis and using household questionnaires from JJ Town in W District, Shanghai, China, we investigate how spatial justice affects residents' policy acceptance of CLR through an ordered probit model. The results show that (1) spatial justice strengthens residents' policy acceptance of CLR; (2) both policy familiarity and participation are important influencing factors that contribute to residents' policy acceptance of CLR; (3) age, education, household income, the contracting land scale and household population structure also affect residents' policy acceptance of CLR. (4) Robustness tests support the above findings. Thus, in the process of CLR, it is essential to fully consider the realization of spatial justice to ensure the development of remote suburbs, especially the regions experiencing a net reduction in their construction land.

**Keywords:** construction land reduction; policy acceptance; rural revitalization; spatial justice; survey; China

## 1. Introduction

Land-use allocation has long been an important research field in regional science [1,2]. In China, the contradiction between strict construction land supply restriction and excessive construction land demand is extremely prominent, especially in economically developed regions, such as Shanghai, Beijing, etc. [3]. In September 2015, China proposed "implement total construction land control and reduction management" (*Overall Plan for Ecological Civilization System Reform* issued by the Central Committee of the Communist Party and the State Council of the People's Republic of China. http://www.gov.cn/guowuyuan/2015-09/21/content_2936327.htm, accessed on 15 January 2023), which upgraded construction land reduction (CLR) to a national strategy. Under the control of the total amount and intensity of construction land, CLR in remote suburbs of economically developed regions has become a policy innovation in land-use and land consolidation. Land consolidation has become a tool for improving the efficiency of land cultivation and supporting rural development [4–6]. To be specific, CLR is a means of land consolidation; it produces a balanced quota of cultivated land occupation and compensation, and it also creates the

development space for new construction land [3]. In essence, it is a transfer of the right to develop [3,7]. China is experiencing unprecedented urbanization [8,9], and construction land has become a tough issue restricting economic development [10]. The scarcity of land resources in Shanghai, China, is rather serious, and the construction land scale is reaching the "ceiling" of urban planning [11].

Land policy is a type of public policy pertaining to space [12]. Shanghai should "strictly control the new construction land" and "adhere to the negative growth of the total planned construction land"; by 2035, "the total scale of construction land shall not exceed 3200 square kilometers" (*Approval of the State Council on the General Urban Planning of Shanghai* (State Letter [2017] No.147). http://www.gov.cn/zhengce/content/2017-12/25/content_5250134.htm, accessed on 15 January 2023). However, by the end of 2014, the total amount of construction land in Shanghai, China, had exceeded 3100 square kilometers, close to the "ceiling" (Shanghai proposes "negative growth" in planning construction land scale. http://finance.people.com.cn/n/2015/1012/c1004-27685348.html, accessed on 15 January 2023). Since 2013, Shanghai has begun to explore and implement CLR policy and is the first provincial city within its region of China to do so [11]. CLR is a requirement for economic and social development as well as ecological civilization construction [11], which contributes to promoting the intensive and efficient use of land resources.

In the process of CLR, specific problematics of Shanghai, China, regarding urbanization, ecology and rural development occur. (1) Insufficient land for construction is an important constraint to urbanization. When addressing the tight constraints on the construction land quota, the number of quota allocations has an impact on the regional development potential. Land is the greatest resource and advantage of rural development. Agricultural development can also have far-reaching influences on subsequent economic development and people's welfare. The infinite expansion of built-up regions leads to a recession in agriculture and natural resources [13]. CLR in Shanghai, China, has restricted the disorderly expansion of construction land. (2) The inefficient construction land in the remote suburbs of cities is seriously polluted and inefficient in output, which reduces the quality of the local ecological environment to a certain extent. Through CLR, some of the highly polluting enterprises are shut down, and the freed construction land quota can provide development space for high-quality enterprises. This has a positive effect on improving the local ecological environment. (3) CLR can also produce uneven allocation of construction land quotas, which undermines the development rights of remote suburbs, especially the net reduction regions of construction land. The remote suburbs are the main regions that implement CLR; however, most of the balanced quota generated from CLR are used by the peri-urban regions. Therefore, while improving the efficiency of land resource allocation, CLR may also weaken the development potential of the remote suburbs. (4) In the absence of a sound compensation mechanism, this development leads to a lack of spatial justice for CLR. Soja links justice to other broad concepts referring to the qualities of a just society, such as freedom, liberty, equality, democracy, civil rights, etc. [14]. Since Rawl's concept of distributional justice [15], the distribution of resources has been central also to the geographical aspect of justice [16]. The development potential of the net reduction regions may be blunted if the administrative regions cannot retain the balanced quota for development, or if the compensation to the net reduction regions is not sufficient. Though the land use regulations could enhance welfare if they generate net benefits for a region or nation [2], they will also restrict the development of vulnerable regions, creating the lack of spatial justice. This implies that the spatial justice of CLR cannot be effectively realized. In this study, spatial justice means that, according to the principle of equity, the reduction obligations of the subject are equivalent to the obligations of other subjects, or the rights to use the balanced quota obtained from CLR for new construction projects are equivalent. However, at present, in the link of task allocation within the process of CLR, the low efficiency of construction land in the remote suburbs make them the key regions of CLR. In the link of balanced quota use, meanwhile, location advantages mean that the peri-urban regions are often allocated a more balanced quota of construction land. As

a result, it is difficult to give full play to the advantage of backwardness of the remote suburbs and protect their development rights and interests, which this study calls the lack of spatial justice.

Spatial justice is crucial to the sustainable development of a region. Sustainable development of suburban regions in economically developed regions is a complex issue, involving urban development, industrialization, problematics of rural villages and the relationship between them. The sustainable development goals reflect the pursuit of achieving spatial justice [17]. What is certain is that sustainable development is not a matter of sacrificing or limiting the development of one region in exchange for the rapid development of another; instead, it should be devoted to inter-regional coordination and co-development. Thus, this study takes CLR as an example to analyze how spatial justice affects residents' support for CLR policies from the perspective of the inequitable allocation of construction land due to CLR. The more the residents support CLR policy, the less resistance the cadres will have for carrying out CLR and the better the policy effect of CLR will be. However, there are few studies on spatial justice and residents' policy acceptance in the context of CLR. This study investigates how spatial justice affects residents' policy acceptance of CLR from both theoretical and empirical perspectives. CLR limited the development of net reduction regions of CLR [3]. This study has policy reference value for improving the CLR and promoting the development of remote suburbs, especially the net reduction regions of CLR.

The contributions of this study are as follows. (1) It theoretically analyzes the connotation of spatial justice in CLR and its influence on residents' policy acceptance of CLR, which can provide a reference for improving residents' acceptance of national economic policy. (2) Based on theoretical analysis and using household questionnaires from JJ Town in W District, Shanghai, China, we investigate how spatial justice affects residents' policy acceptance through the ordered probit model, which enriches the quantitative research on spatial justice and residents' policy acceptance. (3) Policy feedback is important [18], but this study discusses other factors that affect residents' policy acceptance, which provides ideas for optimizing the existing policy of CLR.

The remainder of the study is structured as follows. Section 2 presents the literature review; Section 3 discusses the theoretical analysis and research hypothesis; Section 4 provides the research design; Section 5 highlights the empirical analysis; Section 6 presents the main conclusions and policy implications.

## 2. Literature Review

CLR is an exploratory development model used to adapt to the tight constraints of construction land [11,19]. Its essence is the transfer and optimal allocation of construction land quota [3]. The core idea is to form a balanced quota by reducing the area of inefficient construction land outside the centralized construction area, and then use the balanced quota to efficiently develop state-owned land in the centralized construction area. CLR is a requirement for the construction of China's economic, social and ecological civilization, and its fundamental goal is to promote the economical, intensive and efficient use of land [11,19,20]. Through CLR, we can realize the optimization of the utilization structure of construction land and solve the construction land use contradiction [19,21]. Industrial land reduction directly reduces the cost of public management and pollution sources and improves the ecological environment [11]. Some studies also focus on the operation mechanism and location selection of CLR [19]. Several studies have focused on industrial land reduction policy's impacts on town- and village-level interests [11] and the full-life-cycle management mechanisms and policies regarding industrial land [11,22]. Some studies believe that land use regulations could be welfare enhancing if they generate net benefits for a region or nation [2]. Some scholars have also studied residents' selection behavior toward compensation schemes for CLR [3].

The academic world has not yet drawn a clear conclusion on the meaning of spatial justice. The existing research mainly focuses on aspects such as the theoretical connotation



of spatial justice, environmental injustice and spatial justice in planning. (1) The relevant research on spatial justice is mainly qualitative research. The concept of spatial justice first appeared in the academic debate in the 1970s [23]. Spatial justice is not about the allocation of equal resources, but about opportunities and capabilities [16]. From the perspective of urban land management, spatial justice can give different categories of urban residents, including the poor and low-income groups, equal opportunities to own and use land [23,24]. (2) Studies generally view environmental injustice as an unequal distribution of burdens, hazards and risks related to environmental policies, plans and projects [25,26]. People call for environmental justice in many cases, such as when irreversible changes occur in the quality or use value of the environment in which people live, when the use of public property resources is affected, when certain groups are not considered or given fair benefits, or when changes in the control or use of the land limits people's ability to fulfil the land's full potential [27,28]. The field of environmental justice research initially emerged in the late 1980s and focused mainly on the disproportionate environmental burden posed by land use activities on certain ethnic, vulnerable and marginalized groups in rich countries [27]. Subsequently, it expanded to gender [29], the global environmental justice movement [30] and the hierarchies and inequalities within vulnerable groups [31]. There are also studies which examined green transformations in Vietnam's energy sector from an energy justice perspective [32]. (3) Spatial justice in planning. Justice is one of the main goals of planning thought [33]. Spatial equity is an important part of sustainable urban planning [34]. Planning has continuity and, in most cases, may exacerbate economic inequality [35]. Justice is one of the main goals of planning thought, and it is also a goal widely accepted by scholars in all periods [33]. The existence and implementation of land use planning, urban planning and land spatial planning may lead to spatial injustice, such as the lack of spatial justice in urban public open space planning [36,37]. Planning standards that are applied across a territory could become a tool for producing spatial injustice under certain circumstances [38]. In order to make a more just contribution to urban society, planning needs to pay attention not only to the distribution mode, but also to the social structure and institutional background [39].

There is little research on residents' policy acceptance, which is the premise and foundation of effective policy implementation. Policy design has returned as a central topic in public policy research [40]. Some research focuses on the role of landscape identification in policy implementation and suggests that it benefits policy implementation by improving people's satisfaction with the surrounding landscape through positive activities [41]. Some studies investigate the problem of farmers' policy satisfaction with homestead withdrawal and find that economic status and perceived policy value can promote homestead policy satisfaction [42]. Some scholars have also paid attention to farmers' satisfaction with the increase–decrease linking policy and found that farmers' willingness to participate, relevant policy knowledge, living conditions before resettlement and resettlement compensation have a significant impact on satisfaction with policy implementation. They believe that the government should provide long-term support plans to help farmers change their lifestyle [43]. Some study measured the residential satisfaction of residents in six urban renewal projects in Chongqing, China [44].

In summary, research on CLR mainly focuses on the impact of CLR on the income of different behavioral subjects and the mechanism of CLR. There are few quantitative studies on spatial justice and how it affects residents' policy acceptance of CLR. (1) The existing literature generally suggests that CLR is the inevitable result of economic development to a certain level. With the rapid development of urbanization, the incremental potential of construction land has decreased sharply. Therefore, it is necessary to reduce construction land, revitalize existing construction land, and realize the more intensive, efficient and sustainable high-quality use of land resources. (2) There are few studies that discuss the spatial justice in CLR. The limited number of cities that have implemented the CLR in China and the lack of data in this respect result in little empirical research. Since the reform and opening up of China, China's land space planning has become more mature

and perfected and has paid increasing attention to fairness, justice and the ecological environment. How does spatial justice affect residents' policy acceptance? How can the residents' policy acceptance be improved under this influence? This study considers the example of Shanghai, which took the lead in the implementation of CLR in China, and conducts research based on questionnaires from JJ Town in W District, Shanghai, China. The influence of spatial justice on residents' policy acceptance as well as factors affecting residents' policy acceptance are thoroughly investigated, and some policy implications are proposed to enhance the existing spatial justice of the current policy of CLR to improve residents' policy acceptance.

This paper provides a policy reference for Shanghai and other economically developed regions to improve their construction land policies and enhance residents' policy acceptance.

## 3. Theoretical Analysis and Research Hypothesis

### 3.1. Construction Land Reduction

CLR is based on the ownership unit of the land. China's land is divided into state-owned and collectively owned land. The latter is further divided into villager-group-owned, village-collective-owned and township-collective-owned land. Before and after CLR, the ownership of the land does not change; only the use of the land is changed, from construction land to agricultural or unused land. Therefore, the basic unit of CLR should be the villager group (production team) collective. Currently, some villager groups have become weakened and the village collective has become more materialized, so the latter can also be regarded as the basic unit of CLR. Some reduced construction land belongs to the township collective, so its unit is the township collective. CLR is implemented with the villager group as the basic unit. In the process of reducing the decentralized construction land in the villager groups to concentrate on the industrial park, government compensation is also carried out by the unit of the villager group. Therefore, this study takes the villager group as the basic research unit.

CLR involves two aspects: task allocation and balanced quota allocation. (1) In the CLR process, the tasks of CLR are decomposed from top to bottom (municipal government→district governments→township governments→underlying subjects) [3]. Regions with poor locations that have more inefficient construction land are the key regions for CLR. Regions with more inefficient construction land (remote suburbs) take more reduction tasks, while regions with less inefficient construction land (peri-urban regions) take fewer reduction tasks. (2) In CLR balanced quota allocation, allocating the construction land quota to regions with better locations will generate more benefits. In standard economic models and theories of economic geography, location is a critical component which affects the productivity and profitability of firms and industries [45–47]. On the basis of the construction land's output efficiency, the government gives priority to the allocation of balanced quota to the peri-urban regions and provides fewer quota to the remote suburbs. Regions with good locations are often experiencing a net incremental in their construction land area, while regions with poor locations are often experiencing a net reduction in their construction land area.

For convenience in analyzing the problem, we assume that there are two types of regions, *A* and *B*. *A* refers to the peri-urban regions with a relatively high construction land productivity. In the process of CLR, construction land quotas will be used for the introduction of new industries, industrial upgrading and structural optimization as well as high-quality development [3]. Construction land productivity reflects the level of possible production for the construction land under certain conditions, which includes both the level of total output and output efficiency. Output efficiency can be measured by the level of production per unit of construction land area. *B* refers to the remote suburbs with relatively low construction land productivity. CLR is a reallocation of land development rights under planning constraints [3]. The basic problem with the development of peri-urban regions and remote suburbs is the transfer of land development rights across regions. This basis allows for the creation of a lack of spatial justice. The remote suburbs are inherently

slow to develop, and CLR further limits the development of these regions, thus making re-urbanization less distinctive.

There are two core points in CLR: first, the total amount of construction land is controlled, that is, the total amount of construction land does not increase; second, the efficiency of construction land does not decrease, which can be formulated as (1) and (2):

$$\sum_{i=1}^{2} \sum_{t=1}^{n} S_{i,t} \leq a \tag{1}$$

$$\sum_{i=1}^{2} \sum_{t=1}^{n} w_{i,t} \cdot S_{i,t} \geq \sum_{i=1}^{2} \sum_{t=1}^{n} w_{i,t-1} \cdot S_{i,t-1} \tag{2}$$

which satisfies the following formula:

$$w_{i,t} \geq w_{i,t-1} \tag{3}$$

$$w_A \geq w_B \tag{4}$$

In Equation (1), *a* is the total area of construction land, referring to the control of the total amount of urban construction land area. *t* refers to time, *i* = 1 refers to region *A*. *i* = 2 refers to region *B*. $S_{i,t}$ refers to the total construction land area of region *i* at time *t*. In Equation (2), $w_{i,t}$ refers to the efficiency of construction land in region *i* at time *t*. Equation (2) means that the total output of construction land does not decrease. Equation (3) means that the efficiency of construction land does not decrease. Equation (4) means that the allocation (efficiency) of construction land in region *A* is better than that in region *B*.

### 3.2. Spatial Justice and its Influence on Resident's Policy Acceptance of CLR

Resource allocation is the choice of a balanced schemes for efficiency and fairness. From the perspective of social rights and interests, the key to fairness is the equality of social rights and interests. Spatial justice should construct a value criterion to realize fairness and rationality as well as the equality of obligations and rights concerning the distribution of spatial resources and products of different groups and regions. The allocation of construction land quota needs to take both efficiency and fairness into account. We should not only consider efficiency factors such as regional resource endowment and differences in economic and social development but also consider the development opportunities of underdeveloped regions. The slow development of the net reduction regions can be attributed to two reasons: on the one hand, there are many reduction tasks in the net reduction regions, which directly weaken the development of those regions; on the other hand, thequota for new construction land that is being allocated to those regions is smaller, which further restricts their development. In terms of balanced quota allocation, it is necessary to fully consider the differences in the level of economic development. For regions with a low level of economic development, it is essential to start from the perspective of fairness, ensure the basic development needs of such regions and guarantee the fairness of the quota allocation. From the viewpoint of the location selection of balanced quota, allocating more balanced quota to the net reduction regions can improve the spatial justice of CLR, which is also in line with the requirement of rural revitalization [1]. In other words, "what is taken from here is used here".

There are two paths to realize the interests of the balanced quota of CLR. (1) For the in situ realization path, the balanced quota generated from CLR are used within the scope of ownership subjects, and the beneficiary groups remain unchanged. (2) The off-site realization path aims to achieve the Pareto improvement of social welfare by realizing the allocation of the balanced quota among different ownership subjects and by realizing the development of the whole region under the premise of ensuring that the development of the net reduction regions does not decrease. In this realization path, the development of net reduction regions needs to be emphasized. If not done well, the development of the net reduction regions will be limited, which is not conducive to rural revitalization and regional balanced development.

In terms of in situ realization, villager groups develop independently, which will contribute to the local economic development, even though the construction land is scattered. There are problems of economic scale, and the industrial agglomeration effect is difficult to implement. For scattered industrial land, the cost of ecological and environmental protection is very high [3,19], which is not conducive to centralized management and does not conform to the development law of urbanization. In the off-site path, an agglomeration economy and scale economy can be realized through industrial agglomeration. If the industrial structure can be optimized, the plot ratio improves, the output of unit construction land area increases accordingly, and the cost of environmental governance and industrial management is greatly reduced. If the net reduction regions can receive sufficient compensation from the quota use regions, whether in the form of technical help or financial support, it will have a catalytic effect on the development of the net reduction regions. The financial support helps to improve both the ecological environment of the net reduction regions and the living conditions of the residents by promoting centralized residence, which can create conditions for better development of the net reduction regions.

In summary, when the balanced quotas are all used within the scope of ownership subjects, it is conducive to the realization of the interests of balanced quota and belongs to spatial justice. If the quota is allocated to other regions crossing among different ownership subjects instead of being used in the local administrative regions, it also contributes to the development of the reduced regions and belongs to spatial justice when the net reduction regions obtain sufficient compensation. If the net reduction regions cannot obtain enough compensation, it is not conducive to their development, which fails to create spatial justice. In the process of CLR, more consideration of the fairness of regional development can contribute to the coordinated win–win development of both the quota outflow and inflow regions. For the allocation of CLR quota, regions with different development levels should be allocated relatively equal balanced quota, especially for underdeveloped regions, and basic development rights should be guaranteed. It is not wise to sacrifice the development of remote suburbs to achieve the development of peri-urban regions. If more quotas are used for the development of the region after CLR, it will help to realize the right of construction land in the region and enhance residents' acceptance of the policy. The following hypothesis is proposed:

**Hypothesis 1 (H1).** *Spatial justice can improve residents' policy acceptance of CLR.*

## 4. Research Design

### 4.1. Model Building

This study focuses on the impact of spatial justice on residents' policy acceptance. Since the dependent variable, residents' policy acceptance, is ordered data, ordinary least square (OLS) might not be applicable. The ordered probit model [48,49] is the extended probit model, which is specifically used to handle estimation problems when the dependent variable is ordered data. The widely used ordered probit model is used for the estimation. The model is as below:

$$RPA_i = F(\beta \cdot SJCLR_i + \gamma \cdot X_i + \varepsilon_i) \tag{5}$$

In Equation (5), $RPA_i$ refers to the dependent variable, representing the CLR policy acceptance of resident $i$. $SJCLR_i$ is the core explanatory variable, representing the spatial justice of CLR of resident $i$. $X_i$ refers to other factors that influence residents' policy acceptance, as well as control variables that reflect the individual characteristics and household characteristics of resident $i$. $F(\bullet)$ is a nonlinear function expressed as follows:

$$F(RPA_i^*) = \begin{cases} 1 & RPA_i^* \leq \mu_1 \\ 2 & \mu_1 < RPA_i^* \leq \mu_2 \\ \vdots & \vdots \\ J & RPA_i^* > \mu_{J-1} \end{cases} \tag{6}$$

In Equation (6), $\mu_1 < \mu_2 < \mu_3 < \ldots < \mu_{J-1}$ is called the point of tangency, all of which are parameters to be estimated. $RPA_i^*$ is the unobservable continuous variable existing behind $RPA_i$, which is called the latent variable and requires the following condition to be satisfied:

$$RPA_i^* = \beta \cdot SJCLR_i + \gamma \cdot X_i + \varepsilon_i \tag{7}$$

### 4.2. Selection of Variables and Indicator Measures

#### 4.2.1. Dependent Variable

Residents' policy acceptance is regarded as the dependent variable. Policy identity plays a very important role in the process of policy implementation [50]. Residents' policy acceptance refers to the sense that a certain public policy in social life belongs to the public and is the tendency of social members to support and approve of a certain policy. Subjective evaluation is taken as a measure of residents' policy acceptance in this study. Residents' policy acceptance is measured using residents' suggestions for the strength of implementation of CLR policies in their town's "14th Five-Year Plan". The "14th Five-Year Plan" plotted the main goals of economic and social development for the period 2021–2025. Moreover, the Plan is strongly future-oriented and needs to be oriented to a longer-term future, not just the next five years. The indicator is a judgment that integrates complex information which can comprehensively reflect residents' degree of policy recognition of CLR.

#### 4.2.2. Explanatory Variable

Spatial justice is the core explanatory variable of this study, using "(Total construction land in 2018—Total construction land in 2013)/Total population in 2013" as a measurement. The larger the value, the higher the degree of spatial justice that is achieved; the smaller the value, the lower the degree of spatial justice that is achieved. This measurement can represent the degree of spatial rights realization of a village caused by CLR policy. The change trend of construction land use area per capita and per land in each village is a reflection of construction land space. The fundamental basis is that, under the system of strict control of the total amount and intensity of construction land in China, the space for construction land is extremely limited. Village with a larger construction land quota will have more development space. Therefore, in villages with small amounts of land used for construction, development space is compressed and rights and interests are damaged. It should be noted that according to the *Master Plan and General Land Use Plan of W District, Shanghai (2017–2035)*, the current construction land area of JJ Town is 9 square kilometers, and the planned construction land area by 2035 is 6.8 square kilometers, which means that from the planning base year (the end of 2016) to 2035, the construction land area of JJ Town will decrease by 15 percentage points. However, this does not mean that all villages in JJ Town are net reduction regions of construction land. Using this method, 13 administrative villages were divided into four groups: the spatial justice low realization group (GM Village, ZM Village, JY Village), medium realization group (JJ Village Committee, ZF Village, ZL Village, NL Village), higher realization group (ST Village, WC Village, YH Village) and highest realization group (ZH Village Committee, YG Village, NT Village).

Since the administrative area of each village is different, in addition to using the per capita construction land area to describe the construction land quantity of each village, we also use the construction land quantity of unit administrative area to reflect the construction land rights of a village. The larger the construction land area per unit administrative area is, the more production and living functions it carries, and the more the development rights and interests of residents can be guaranteed. Therefore, the village has advantages in development space and belongs to the beneficiary of development space. However, in villages with a small unit administrative area, the development space is small, and the rights and interests of residents in production and life are damaged, which results in a lack of spatial justice of CLR. Therefore, this study re-measures spatial justice from the perspective of construction land area of unit administrative division area. In the robustness

test, we also use "(Total construction land in 2018–Total construction land in 2013)/Total construction land in 2013" to measure spatial justice of CLR.

In practice, the reduction and use of quota often cannot be accurately matched. That is, the net reduction regions are often remote suburbs or relatively underdeveloped regions, while the balanced quota use regions are often the peri-urban regions or regions with relatively good economic development (peri-urban regions, government regions or regions near the government regions of the remote suburbs). Concerning the quota trading radius, if the quota obtained by CLR can be used for the development of the net reduction regions, then this theoretically guarantees the development of CLR regions to a certain extent. The smaller the trade radius (that is, for all of the township), the stronger the spatial justice of CLR. Thus, from the perspective of the quota trading radius, we also use "How should the CLR quota obtained in the implementation process of the town's policy of CLR be used?" to measure spatial justice of CLR.

### 4.2.3. Control Variables

The control variables include three aspects: (1) policy variables affecting residents' policy acceptance, including policy familiarity (*PF*), compensation standard (*CS*) and policy participation (*PP*). Landowners' familiarity with land renovation helps to explain landowners' behavioral motivation for land renovation and other related projects [51]. In addition, the compensation standard can affect residents' policy acceptance and was controlled. Policy participation is also an important influencing factor for residents' policy acceptance. The active participation of landowners in land renovation projects helps them to understand policy objectives and express satisfaction with land renovation results, which also affects their familiarity with land renovation projects [51]. Therefore, policy participation is also included in the control variables. Residents mainly participate in policy implementation rather than in policy formulation. (2) Resident's individual characteristics include gender (*GEN*), age (*AGE*), and level of education (*EDU*). (3) Household characteristics mainly include residents' household income in 2019 (*HI*), contracting land scale (*CLS*), and household population structure (*HPS*). The specific interpretation and indicator measures of each variable of the model are shown in Table 1.

### 4.3. Data Source

In 2014, JJ Town, W District of Shanghai took the lead in implementing the policy of CLR to pursue the high-quality use of land. The town has a good foundation for CLR, and the district's policies are relatively systematic and well developed. As a result, the research group selected JJ Town, W District, Shanghai, as the research site according to the economic development conditions and location characteristics, covering all characteristic types of CLR in Shanghai, China. The data collected in this study are not unique. They represent the CLR situation and can be used to evaluate residents' policy acceptance of CLR. At the village level, some villages have a net increase in construction land area, which belongs to the net increment regions, while some villages have a net decrease in construction land area, which belongs to the net reduction regions. This is the geospatial unit and basis of the spatial justice analysis in this study. JJ Town (which adopted the town and district integrated administrative system with W District modern agricultural park) is located in the middle-southwest of W District, Shanghai, China. According to the *2020 Statistical Yearbook of W District*, the registered population of JJ Town was 30,939, with 9082 households and a permanent population of 29,600 in 2019. Among them, 4671 were under the age of 18, 7973 were aged 18–35, 21,773 were aged 35–60, and 16,292 were aged above 60. To analyze the impact of spatial justice of CLR on residents' policy acceptance, the research group went to JJ Town and collected data through interviews and questionnaires. Residents of 11 administrative villages and 2 village committees were visited. A total of 400 questionnaires were distributed by random sampling. After eliminating samples lacking information and anomalous samples, 344 effective questionnaires were collected (306 local resident questionnaires), with an effective recovery rate of 86%. The survey was conducted from

September to October 2020. Construction land data were extracted from remote sensing image data by ArcGIS. Basic information on the survey subjects can be seen in Table 2.

**Table 1.** Description of model variables.

| Variable Types | Variable Name | Measurement |
|---|---|---|
| Dependent variable | Residents' policy acceptance (*RPA*) | Residents' suggestions for the strength of implementation of CLR policies in their Town's "14th Five-Year Plan". Strong acceptance (actively increase the intensity) = 5; partial acceptance (appropriately increase the intensity) = 4; neutral (basically the same as in recent years) = 3; partial unacceptance (appropriately reduce the intensity) = 2; strong unacceptance (strongly reduce the intensity) = 1 |
| Core explanatory variable | Spatial justice of CLR (*SJCLR*$_1$) | (Total construction land in 2018–Total construction land in 2013)/Total population in 2013. >75% percentile = 4; 50–75% percentile = 3; 25–50% percentile = 2; <25% percentile = 1 |
| | Spatial justice of CLR (*SJCLR*$_2$) | (Total construction land in 2018–Total construction land in 2013)/Total construction land in 2013. >75% percentile = 4; 50–75% percentile = 3; 25–50% percentile = 2; <25% percentile = 1 |
| | Spatial justice of CLR (*SJCLR*$_3$) | How should the CLR quota obtained in the implementation process of the town's policy of CLR be used? All for the township = 3; Most used in the township = 2; Others = 1 |
| Policy variables | Policy familiarity (*PF*) | Are you familiar with the planning and policies in this town? Very familiar or relatively familiar = 1; Others = 0 |
| | Compensation standard (*CS*) | What do you think of the rationality of the compensation standard for CLR in this town? Very reasonable or relatively reasonable = 1; Others = 0 |
| | Policy participation (*PP*) | What do you think of the participation of farmers and collectives in the implementation of the town's policy of CLR? Very active or relatively active = 1; Others = 0 |
| Individual characteristics | Gender (*GEN*) | Male = 1; Female = 0 |
| | Age (*AGE*) | 60 and above = 4; 45–60 = 3; 31–45 = 2; 30 and below = 1 |
| | Level of education (*EDU*) | College and above = 4; Upper secondary school = 3; Lower secondary school = 2; Primary school and below = 1 |
| Household characteristics | Household income (*HI*) | 200,000 CNY and above = 4; 100,000 CNY–200,000 CNY = 3; 50,000 CNY–100,000 CNY = 2; 50,000 CNY and below = 1 |
| | Contracting land scale (*CLS*) | 2 mu and above = 2; 0–2 mu = 1; 0 mu = 0 |
| | Household population structure (*HPS*) | The proportion of the population aged 18 to 60 years in the total household size |

Note: 1 mu ≈ 0.0667 hectares.

According to the *2019 Statistical Yearbook of W District*, the urbanization rate of the registered population in W District, Shanghai is 376,169/524,958≈71.66%. For the actual situation, although many residents live in rural regions, they have a nonagricultural household registration. The sample is representative regarding gender composition, age distribution, and educational background. From the perspective of sample size, in 2019, the registered population of JJ Town, W District was 29,600, and the sample size of this study was 306, accounting for about 1.03% of the registered population. The total number of households is 9082, and the investigated households in this study accounted for about 3.37%. The descriptive statistics for each variable can be seen in Table 3.

The distribution of the level of residents' policy acceptance of CLR is shown in Figure 1. The majority of the respondents had a higher degree of policy acceptance (partial acceptance and strong acceptance), while fewer respondents chose neutrality and partial unacceptance. No respondents considered the policy strongly unacceptable. Why did the majority of the respondents have a higher degree of policy acceptance? This may have occurred because the regional ecological environment has been improved through CLR. CLR has also promoted the construction of special rural tourism and idyllic towns. In fact, through CLR, the town has built a country park, which has a positive effect on the local economic development and the level of the residents' well-being.

**Table 2.** Basic information of the survey subjects.

| Item | Component | Freq. | Percent (%) | Cum. (%) |
|---|---|---|---|---|
| *Gender* | Male | 165 | 53.92 | 53.92 |
| | Female | 141 | 46.08 | 100 |
| *Age* | 30 and below | 51 | 16.67 | 16.67 |
| | 31–45 | 104 | 33.99 | 50.66 |
| | 45–60 | 96 | 31.37 | 82.03 |
| | 60 and above | 55 | 17.97 | 100 |
| *Education* | Primary school and below | 42 | 13.73 | 13.73 |
| | Lower secondary school | 108 | 35.29 | 49.02 |
| | Upper secondary school | 59 | 19.28 | 68.30 |
| | College and above | 97 | 31.70 | 100 |
| *Household scale* | 3 and below | 51 | 16.67 | 16.67 |
| | 4 | 55 | 17.97 | 34.64 |
| | 5 | 132 | 43.14 | 77.78 |
| | 6 and above | 68 | 22.22 | 100 |
| *Contracting land scale* | 0 | 106 | 34.64 | 34.64 |
| | 0–2 mu | 92 | 30.07 | 64.71 |
| | 2 mu and above | 108 | 35.29 | 100 |
| *Household income* | 50,000 CNY and below | 47 | 15.36 | 15.36 |
| | 50,000 CNY–100,000 CNY | 77 | 25.16 | 40.52 |
| | 100,000 CNY–200,000 CNY | 124 | 40.52 | 81.05 |
| | 200,000 CNY and above | 58 | 18.95 | 100 |

**Table 3.** Descriptive statistics.

| Variable | Obs | Mean | Std.Dev | Min | Max |
|---|---|---|---|---|---|
| $RPA$ | 306 | 3.9641 | 0.9728 | 2 | 5 |
| $SJCLR_1$ | 306 | 2.5948 | 1.1647 | 1 | 4 |
| $SJCLR_2$ | 306 | 2.5098 | 1.0932 | 1 | 4 |
| $SJCLR_3$ | 306 | 1.9542 | 0.8325 | 1 | 3 |
| $PF$ | 306 | 0.3856 | 0.4875 | 0 | 1 |
| $CS$ | 306 | 0.5098 | 0.5007 | 0 | 1 |
| $PP$ | 306 | 0.4281 | 0.4956 | 0 | 1 |
| $GEN$ | 306 | 0.5392 | 0.4993 | 0 | 1 |
| $AGE$ | 306 | 2.5065 | 0.9726 | 1 | 4 |
| $EDU$ | 306 | 2.6895 | 1.0613 | 1 | 4 |
| $HI$ | 306 | 2.6307 | 0.9603 | 1 | 4 |
| $CLS$ | 306 | 1.0065 | 0.8376 | 0 | 2 |
| $HPS$ | 306 | 0.6380 | 0.2407 | 0 | 1 |

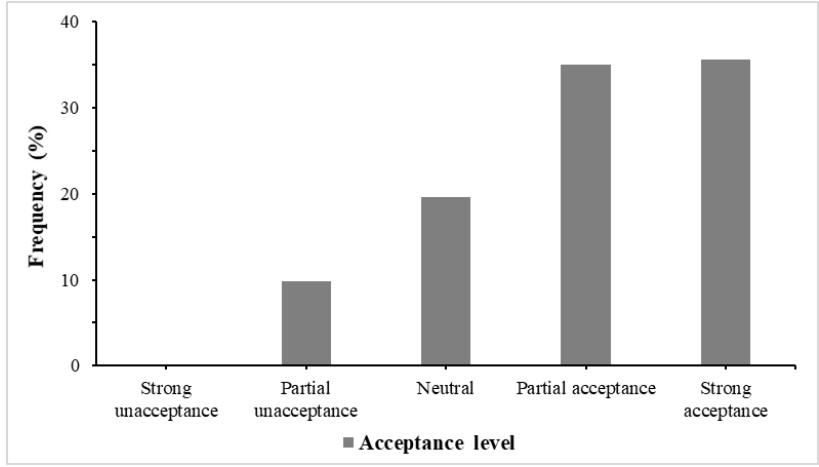

**Figure 1.** Distribution of the level of residents' policy acceptance of CLR.

## 5. Empirical Analysis

### 5.1. Preliminary Data Analysis

The regression results of the ordered probit model are shown in Table 4. Column (1) in Table (4) is the most basic regression and contains only the explanatory variables and constant items of greatest concern in this study without the control variables. Spatial justice of CLR significantly promoted residents' policy acceptance when control variables are not included. The improvement of spatial justice of CLR significantly enhances residents' policy acceptance. In Table 4 Column (2), other factors affecting residents' policy acceptance are controlled, including *PF*, *CS* and *PP*. All the above factors have a significant positive impact on residents' policy acceptance. The stronger policy familiarity is, the stronger residents' policy acceptance is. The more reasonable the compensation standard is, the more it helps to enhance residents' policy acceptance. Moreover, strengthening residents' policy participation can also enhance residents' policy acceptance of CLR. We further controlled the individual characteristics of the residents, including *GEN*, *AGE* and *EDU*. The results are displayed in Table 4, Column (3) and show that gender has no significant impact on residents' policy acceptance, while age and the level of education affect residents' policy acceptance to a great extent. Residents who were older and had more education also tended to have stronger policy acceptance. CLR is a macro policy in line with the law of urbanization and industrial development; the higher respondents' education is, the deeper their understanding of the policy. People with higher education have stronger competitiveness and are less affected by CLR in terms of work, so their policy acceptance is relatively stronger. Residents who are older have stronger residents' policy acceptance because CLR provides funds for the development of villages and towns, which can be used to improve rural elderly activity centers and places. Villages gain money to provide lunches for the elderly, which improves living welfare. Therefore, the older the residents are, the stronger their policy acceptance. In addition, we controlled the household characteristics of residents, mainly including residents' *HI* in 2019, *CLS*, and *HPS*. The results are shown in Table 4, Column (4). The impact of these factors is not significant, but the conclusions still hold. Spatial justice of CLR greatly enhances residents' policy acceptance. With the introduction of the control variables, the pseudo $R^2$ increases from 0.0041 to 0.1172.

**Table 4.** Ordered probit model regression results.

| Variable | (1) | (2) | (3) | (4) |
|---|---|---|---|---|
| $SJCLR_1$ | 0.0955 * | 0.1552 *** | 0.1346 ** | 0.1852 *** |
| | (0.0490) | (0.0501) | (0.0529) | (0.0577) |
| *PF* | | 0.4392 *** | 0.3466 ** | 0.3373 ** |
| | | (0.1474) | (0.1620) | (0.1634) |
| *CS* | | 0.2703 * | 0.2780 * | 0.2361 |
| | | (0.1435) | (0.1477) | (0.1592) |
| *PP* | | 0.4268 *** | 0.4894 *** | 0.5457 *** |
| | | (0.1551) | (0.1612) | (0.1733) |
| *GEN* | | | −0.1577 | −0.1317 |
| | | | (0.1344) | (0.1343) |
| *AGE* | | | 0.3278 *** | 0.3183 *** |
| | | | (0.0934) | (0.0969) |
| *EDU* | | | 0.2930 *** | 0.3020 *** |
| | | | (0.0848) | (0.0815) |
| *HI* | | | | −0.1999 *** |
| | | | | (0.0699) |
| *CLS* | | | | 0.1719 ** |
| | | | | (0.0828) |
| *HPS* | | | | 0.7625 ** |
| | | | | (0.3002) |

**Table 4.** *Cont.*

| Variable | (1) | (2) | (3) | (4) |
|---|---|---|---|---|
| /cut1 | −1.0546 *** | −0.5360 *** | 0.8651 ** | 1.0702 ** |
| | (0.1438) | (0.1670) | (0.4276) | (0.5063) |
| /cut2 | −0.2995 ** | 0.2859 * | 1.7440 *** | 2.0154 *** |
| | (0.1453) | (0.1630) | (0.4364) | (0.5247) |
| /cut3 | 0.6197 *** | 1.2984 *** | 2.7884 *** | 3.0974 *** |
| | (0.1497) | (0.1620) | (0.4420) | (0.5320) |
| Observations | 306 | 306 | 306 | 306 |
| Pseudo $R^2$ | 0.0041 | 0.0678 | 0.0912 | 0.1172 |

Notes: *** $p < 0.01$, ** $p < 0.05$, * $p < 0.1$; Robust standard errors in parentheses.

Since the parameter meaning of the ordered probit model is not intuitive, the results in Table 4 can only give limited information in terms of significance and parametric symbols. Therefore, this study obtains the marginal effect of each explanatory variable on the residents' policy acceptance through further calculation. We calculate the probability of how the unit change of the explanatory variable affects the dependent variable when all the explanatory variables are at the mean values, as in Equation (8):

$$\left.\frac{\partial Prob(RPA = i|x)}{\partial x}\right|_{x=\bar{x}} \quad (i = 1, 2, 3, 4, 5) \tag{8}$$

In Equation (8), $x$ represents all the explanatory variables in the regression model. The meaning of the marginal effect of Equation (8) is how the probability of the dependent variable taking the individual value changes when the explanatory variable changes by one unit. As shown in Table 5, Columns (1)–(4) are probability predictions of partially unacceptance, neutral, partially acceptance, and strongly acceptance. We interpret Table 5 with the example of spatial justice of CLR. When all the explanatory variables are at the mean values, spatial justice of CLR will reduce the probability of residents' partial unacceptance and neutrality and increase the probability of residents' strong acceptance of CLR. Therefore, the enhancement of spatial justice of CLR will help to strengthen residents' policy acceptance.

**Table 5.** Marginal effects of explanatory variables in the ordered probit model.

| | (1) | (2) | (3) | (4) |
|---|---|---|---|---|
| Variable | Partial Unacceptance | Neutral | Partial Acceptance | Strong Acceptance |
| $SJCLR_1$ | −0.0262 *** | −0.0272 *** | −0.0043 | 0.0577 *** |
| | (0.0086) | (0.0088) | (0.0030) | (0.0176) |
| PF | −0.0477 ** | −0.0496 ** | −0.0078 | 0.1050 ** |
| | (0.0233) | (0.0234) | (0.0068) | (0.0502) |
| CS | −0.0334 | −0.0347 | −0.0055 | 0.0735 |
| | (0.0230) | (0.0232) | (0.0052) | (0.0494) |
| PP | −0.0771 *** | −0.0802 *** | −0.0126 | 0.1699 *** |
| | (0.0262) | (0.0266) | (0.0087) | (0.0531) |
| GEN | 0.0186 | 0.0194 | 0.0030 | −0.0410 |
| | (0.0191) | (0.0197) | (0.0036) | (0.0416) |
| AGE | −0.0450 *** | −0.0468 *** | −0.0073 | 0.0991 *** |
| | (0.0144) | (0.0144) | (0.0051) | (0.0290) |
| EDU | −0.0427 *** | −0.0444 *** | −0.0070 | 0.0941 *** |
| | (0.0120) | (0.0128) | (0.0049) | (0.0247) |
| HI | 0.0282 *** | 0.0294 *** | 0.0046 | −0.0623 *** |
| | (0.0104) | (0.0102) | (0.0036) | (0.0216) |

**Table 5.** *Cont.*

| Variable | (1) Partial Unacceptance | (2) Neutral | (3) Partial Acceptance | (4) Strong Acceptance |
|---|---|---|---|---|
| CLS | −0.0243 ** (0.0122) | −0.0253 ** (0.0122) | −0.0040 (0.0032) | 0.0535 ** (0.0257) |
| HPS | −0.1077 ** (0.0433) | −0.1121 ** (0.0452) | −0.0176 (0.0126) | 0.2374 *** (0.0908) |
| Observations | 306 | 306 | 306 | 306 |

Notes: *** $p < 0.01$, ** $p < 0.05$, * $p < 0.1$; Standard errors in parentheses.

## 5.2. Robustness Test

Based on the preliminary data analysis, we tried to test the reliability of the research conclusions with different methods. Residents' policy acceptance was adopted as the dependent variable, and $SJCLR_2$ and $SJCLR_3$ were adopted as the core explanatory variables. The marginal effects of the explanatory variables are shown in Table 6. In Table 6, Columns (1)–(4) are probability predictions of partially unacceptance, neutral, partially acceptance, and strongly acceptance, respectively. Columns (5)–(8) are probability predictions of partial unacceptance, neutral, partial acceptance, and strong acceptance, respectively. When all the explanatory variables are at the mean value, the spatial justice of CLR will reduce the probability of residents' partial unacceptance and neutrality and increase the probability of residents' strong acceptance of CLR. Therefore, the enhancement of spatial justice of CLR will help to strengthen residents' policy acceptance.

**Table 6.** Marginal effects of explanatory variables in the ordered probit model. (Transformation core explanatory variables).

| Variable | (1) Partial Un-acceptance | (2) Neutral | (3) Partial Acceptance | (4) Strong Acceptance | (5) Partial Un-acceptance | (6) Neutral | (7) Partial Acceptance | (8) Strong Acceptance |
|---|---|---|---|---|---|---|---|---|
| $SJCLR_2$ | −0.0221 ** (0.0091) | −0.0228 ** (0.0095) | −0.0034 (0.0027) | 0.0483 ** (0.0192) | | | | |
| $SJCLE_3$ | | | | | −0.0687 *** (0.0128) | −0.0684 *** (0.0129) | −0.0141 * (0.0074) | 0.1511 *** (0.0242) |
| PF | −0.0456 * (0.0235) | −0.0472 ** (0.0236) | −0.0070 (0.0065) | 0.0999 ** (0.0505) | −0.0380 * (0.0221) | −0.0378 * (0.0212) | −0.0078 (0.0060) | 0.0836 * (0.0471) |
| CS | −0.0318 (0.0231) | −0.0329 (0.0233) | −0.0049 (0.0049) | 0.0696 (0.0495) | −0.0141 (0.0230) | −0.0140 (0.0229) | −0.0029 (0.0051) | 0.0310 (0.0507) |
| PP | −0.0797 *** (0.0265) | −0.0824 *** (0.0268) | −0.0123 (0.0089) | 0.1744 *** (0.0534) | −0.0680 *** (0.0255) | −0.0677 *** (0.0246) | −0.0139 * (0.0077) | 0.1496 *** (0.0518) |
| GEN | 0.0175 (0.0193) | 0.0182 (0.0198) | 0.0027 (0.0034) | −0.0384 (0.0418) | 0.0243 (0.0182) | 0.0243 (0.0182) | 0.0050 (0.0046) | −0.0536 (0.0399) |
| AGE | −0.0456 *** (0.0147) | −0.0472 *** (0.0145) | −0.0070 (0.0051) | 0.0998 *** (0.0291) | −0.0344 ** (0.0135) | −0.0343 *** (0.0132) | −0.0070 (0.0043) | 0.0757 *** (0.0284) |
| EDU | −0.0430 *** (0.0122) | −0.0445 *** (0.0129) | −0.0066 (0.0049) | 0.0942 *** (0.0251) | −0.0354 *** (0.0114) | −0.0352 *** (0.0119) | −0.0072 * (0.0044) | 0.0778 *** (0.0246) |
| HI | 0.0240 ** (0.0102) | 0.0248 ** (0.0100) | 0.0037 (0.0031) | −0.0525 ** (0.0212) | 0.0144 (0.0095) | 0.0144 (0.0092) | 0.0030 (0.0024) | −0.0317 (0.0203) |
| CLS | −0.0240 * (0.0123) | −0.0248 ** (0.0122) | −0.0037 (0.0031) | 0.0524 ** (0.0256) | −0.0151 (0.0110) | −0.0150 (0.0109) | −0.0031 (0.0027) | 0.0331 (0.0240) |
| HPS | −0.1040 ** (0.0435) | −0.1076 ** (0.0450) | −0.0160 (0.0120) | 0.2276 ** (0.0905) | −0.0945 ** (0.0398) | −0.0941 ** (0.0392) | −0.0193 * (0.0115) | 0.2079 ** (0.0832) |
| Observations | 306 | 306 | 306 | 306 | 306 | 306 | 306 | 306 |

Notes: *** $p < 0.01$, ** $p < 0.05$, * $p < 0.1$; Standard errors in parentheses.

In terms of the research methods, we also performed a robustness test using the probit model and OLS [52] by reassigning virtual variables of 0 and 1 to the dependent variables to re-estimate with the probit model. The results are shown in Table 7. The coefficients are the marginal effects of the explanatory variables. In Columns (1)–(3), the core explanatory variables are $SJCLR_1$, $SJCLR_2$, and $SJCLR_3$. It can be found that spatial justice of CLR significantly increases residents' policy acceptance.

**Table 7.** Probit model results (marginal effects) and OLS regression results.

| Variable | (1) Partial Acceptance and Strong Acceptance | (2) Partial Acceptance and Strong Acceptance | (3) Partial Acceptance and Strong Acceptance | (4) z_RPA | (5) z_RPA | (6) z_RPA |
|---|---|---|---|---|---|---|
| $SJCLR_1$ | 0.0782 *** (0.0199) | | | 0.1554 *** (0.0463) | | |
| $SJCLR_2$ | | 0.0742 *** (0.0221) | | | 0.1289 *** (0.0492) | |
| $SJCLR_3$ | | | 0.1265 *** (0.0270) | | | 0.3846 *** (0.0647) |
| PF | −0.0588 (0.0581) | −0.0670 (0.0588) | −0.0717 (0.0570) | 0.2256 * (0.1256) | 0.2101 * (0.1264) | 0.1609 (0.1181) |
| CS | 0.0308 (0.0620) | 0.0292 (0.0618) | −0.0073 (0.0617) | 0.1921 (0.1310) | 0.1812 (0.1318) | 0.1034 (0.1339) |
| PP | 0.2657 *** (0.0667) | 0.2721 *** (0.0667) | 0.2221 *** (0.0659) | 0.4669 *** (0.1372) | 0.4823 *** (0.1391) | 0.3981 *** (0.1354) |
| GEN | −0.0420 (0.0460) | −0.0418 (0.0465) | −0.0482 (0.0458) | −0.1052 (0.1066) | −0.1007 (0.1076) | −0.1358 (0.1018) |
| AGE | 0.1081 *** (0.0294) | 0.1111 *** (0.0298) | 0.0933 *** (0.0290) | 0.2696 *** (0.0760) | 0.2715 *** (0.0764) | 0.2138 *** (0.0732) |
| EDU | 0.1284 *** (0.0257) | 0.1280 *** (0.0261) | 0.1145 *** (0.0259) | 0.2625 *** (0.0640) | 0.2642 *** (0.0652) | 0.2286 *** (0.0618) |
| HI | −0.0494 * (0.0265) | −0.0373 (0.0263) | −0.0193 (0.0255) | −0.1712 *** (0.0565) | −0.1440 *** (0.0554) | −0.0886 * (0.0528) |
| CLS | 0.0629 ** (0.0291) | 0.0628 ** (0.0293) | 0.0346 (0.0269) | 0.1500 ** (0.0671) | 0.1470 ** (0.0670) | 0.0966 (0.0624) |
| HPS | 0.3378 *** (0.1017) | 0.3279 *** (0.1035) | 0.3141 *** (0.0974) | 0.6578 *** (0.2315) | 0.6242 *** (0.2308) | 0.5691 *** (0.2115) |
| Constant | | | | −2.2559 *** (0.3998) | −2.2300 *** (0.4037) | −2.3639 *** (0.3700) |
| Observations | 306 | 306 | 306 | 306 | 306 | 306 |
| R-squared | | | | 0.2736 | 0.2633 | 0.3302 |

Notes: *** $p < 0.01$, ** $p < 0.05$, * $p < 0.1$; Standard errors in parentheses in Columns (1)–(3); Robust standard errors in parentheses in Columns (4)–(6).

Furthermore, we conducted a robustness test using OLS [52], carrying out regression analysis to *z_RPA* standardized from residents' policy acceptance. The regression results are displayed in Table 7. In Columns (4)–(6), the core explanatory variables are $SJCLR_1$, $SJCLR_2$, and $SJCLR_3$. It can be found that spatial justice of CLR significantly promotes the residents' policy acceptance of CLR. In conclusion, regardless of whether the core explanatory variable or the estimation method is changed, the research conclusion is robust.

## 6. Conclusions and Policy Implications

### 6.1. Conclusions

As an important strategic resource, the construction land quota is very important to regional economic development. CLR has an impact on justice while improving the efficiency of resource allocation. Strengthening the spatial justice of CLR is of practical significance for solving the contradiction between unbalanced and inadequate development and people's ever-growing needs for a better life. Based on the micro household questionnaire survey

from JJ Town in W District, Shanghai, China, this study investigated the influence of spatial justice of CLR on residents' policy acceptance with the ordered probit model. The results showed that spatial justice of CLR plays a significant role in promoting residents' policy acceptance. The research hypothesis of this study was tested.

The main conclusions are as follows: (1) spatial justice significantly strengthens the degree of residents' policy acceptance of CLR. (2) Both policy familiarity and policy participation are important influencing factors contributing to residents' policy acceptance. (3) From an individual characteristics perspective, old people are more concerned about spatial justice than young people. People with high education are more concerned about spatial justice than people with low education. (4) From a household characteristics perspective, household income has a negative impact on residents' policy acceptance, while contracting land scale and household population structure have positive effects on residents' policy acceptance. (5) The robustness tests, including variable remeasurement (adjusting the core explanatory variables) and changing estimation methods, all support the research conclusions of this study. Thus, enhancing the spatial justice of CLR is of positive significance for enhancing the effect of CLR policy from the perspective of residents.

### 6.2. Policy Implications

Because the distribution of construction land quotas in the CLR process is uneven, the rights and obligations of regions for quota reduction and quota use are not equal based on the principle of equity, which is a typical feature of the lack of spatial justice. The lack of spatial justice will weaken residents' policy acceptance of CLR. Improving residents' CLR policy acceptance not only helps reduce the resistance to its implementation, but also helps enhance the effectiveness of the implementation of CLR. In the context of rural revitalization strategy, the contradiction between supply and demand of construction land will become more prominent. In future urban development processes, CLR policy will become increasingly difficult to implement, but should continue to be implemented [3]. In order to continuously promote the implementation of CLR, residents' policy acceptance of CLR is an important consideration. Therefore, strengthening the spatial justice of CLR is of great practical significance to promote the sustainable development of cities, especially the net reduction regions of CLR.

The conclusions of this study have the following policy implications. (1) The maintenance of the spatial justice of CLR is conducive to protecting the interests of the net reduction regions of construction land. The lack of spatial justice affects the development of the remote suburbs, especially the net reduction regions of CLR. Against the background of rural revitalization, the development of remote suburbs requires more construction land quota. The use of CLR quota in peri-urban regions goes against the development of remote suburbs and reduces the potential for rural revitalization. Therefore, it is particularly important to strengthen the spatial justice of CLR. (2) In the process of urbanization, attention should be paid to the development of areas with net reduction of construction land. It is necessary both to improve the existing CLR policies and to strengthen the policy inclination for remote suburbs during future urban planning and regulation making. With regard to specific policies, on the one hand, the balanced quota of CLR should be reserved more for the development of the net reduction regions. The remote suburbs should optimize the spatial allocation of construction land, improve the efficiency of land use and enhance the ability to realize the interests of the region. On the other hand, while optimizing CLR quotas in each region, the government should formulate regulatory policies to increase compensation intensity and expand the scope of compensation to protect the development rights of the net reduction regions of construction land.

The remote suburbs are "important battlefields" for CLR. Although allocating all the quota to the reduction regions can achieve absolute justice of CLR, it is not always optimal for the whole region and the net reduction region itself. Because of the limited efficiency improvement space, cross-regional quota allocation is inevitable. In this case, there are two ways to achieve spatial justice of CLR. The first is to increase the compensation for

the transfer quota, so that the net reduction regions will have more funds for regional economic development. Second, residents of the net reduction regions should move with the transfer of balanced quotas to ensure that these people enjoy the benefits of the net increment regions. These benefits include better education, health care, and employment conditions in the net increment regions. At the same time, as the number of residents in the net reduction regions decreases, those who remain in the net reduction region also enjoy more development benefits per capita, thus ensuring the realization of spatial justice of CLR.

**Author Contributions:** Conceptualization, K.W., J.L. and H.L.; methodology, K.W., J.L., H.L., F.Y., F.D. and X.Z.; investigation, K.W., J.L., H.L., F.Y., F.D. and X.Z.; writing—original draft preparation, K.W., J.L. and H.L.; writing—review and editing, K.W., J.L. and H.L.; funding acquisition, K.W. and H.L. All authors have read and agreed to the published version of the manuscript.

**Funding:** This work was supported by the National Office for Philosophy and Social Science of China, grant number 22AGL027; the Shanghai Planning Office of Philosophy and Social Science, grant number 2020BJB010; the Technology Innovation Center for Land Spatial Eco-restoration in the Metropoli-tan Area, MNR, Shanghai, 200003, grant number CXZX202201; and the Fundamental Research Funds for the Central Universities, grant number 2022110229.

**Institutional Review Board Statement:** Not applicable.

**Informed Consent Statement:** Not applicable.

**Data Availability Statement:** Not applicable.

**Conflicts of Interest:** The authors declare no conflict of interest.

## Note

1.  The 19th National Congress of the Communist Party of China first proposed the implementation of the rural revitalization strategy, pointing out that we should adhere to the priority development of agriculture and rural areas, and also stating that we should accelerate the modernization of agriculture and rural areas.

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
