# Peer review of "Spatial Justice and Residents’ Policy Acceptance: Evidence from Construction Land Reduction in Shanghai, China"

_land, doi:10.3390/land12020300_

Round 1

Reviewer 1 Report

* It is not quite clear, which is the relevance of this study. What would be the impact of the results regarding public policy?

* The abstract does not mention many of the problematics presented in the body of the text.

* To begin with, which is the specific problematics of the Shanghai area regarding urbanization, ecology and rural communities. Some maps about the situation would help to understand the Chinese/ Shanghai context.

* How the sustainable development is understood here? How the economical development is understood? I can distinguish different areas of interest here: urban development, industrialization, problematics of rural villages and the relationhip between them. But what is concretely the situation in all these cases is not explained. And especially, how is the "spatial justice" understood? The concept has not been clearly explained. And what is the specific Chinese/ Shanghai region situation of "economic inequality", "social structure", "distribution model", "positive activities", "problem of farmers", "living conditions before resettlement and resettlement compensation", "lifestyle", "disintegration and disappearance of traditional villages", "the lack of urban rights", "socialist spatial justice", "problems of unbalanced and inadequate development", "development opportunities of underdeveloped regions", "rural revitalization", "in-place realization path for spatial justice-off-site realization path" and "improvement of the ecological environment and living conditions".

* The villager group as the basic research unit-who are they exactly? From which part of the Shanghai area? Why was this group chosen?

* Construction land for what? How is land productivity unerstood here?

* What is the basic problem with the development of suburbs and peri-urban areas? Are they of ru-urban character?

* Conclusions do not make clear how the research results might improve the spatial justice (response to the hypothesis).

* ‘14th Five-Year’ construction land reduction policy is not explained.

* Why did the majority of the respondents have a higher degree of acceptance of the policy? Answer to that might give clearer guidelines to the future decision making regarding public policy.

* In general, the results are not explained considering the particular Shanghai urban development background. Their relevance for the future urban development is not clear, not their possible impact regarding the future urban planning and regulations.

Author Response

Dear reviewer:
We would like to thank you for the time and effort that you have put into reviewing the previous manuscript. We are very grateful for your insightful and constructive comments about the original manuscript. We have cautiously understood the insightful comments and professional suggestions, and some revisions are made in accordance with your comments. In the revised manuscript, all changes are marked with yellow shading. We have provided a point-by-point response to your constructive comments. Please see the attachment.
Sincerely yours,
Authors

Reviewer 2 Report

REVIEWER COMMENTS

 The paper needs minor revision and the theortical background should be expanded in line with the recent literature in the field of spatial justice.

First of all, I would like to say that this manuscript is a good one with an interesting analysis based on the topics of spatial justice and the current policies in local and regional territorial development. The article covers the topics included in the main subjects of Land Journal and I recommend to be considered for the publication after minor revision.

TITLE

The article’s title is suitable with the content of the paper and the comparative analysis is welcome in line with both the text body and the main findings of the research. The title of the paper

Spatial Justice and Resident’s Policy Acceptance: Evidence 2 from Construction Land Reduction in Shanghai, China. The paper deals with the spatial justice and current policies in local and regional urban development, with this topic being of a large interest in the current academic backgrounds.

ABSTRACT

The abstract is well-designed and briefly express the present research thus being of interests and readable thus capturing the reader’s attention. It present in an appropriate manner the main research hypothesis, the problem statement, the methods and the main findings.

KEY WORDS

The key words are appropriate to the present research and are clearly stated.

ORIGINALITY

The article meets a high level of originality from my side argued by the main research theme and the research hypothesis. Furthermore, the originality of the paper is highlighted by the main results of the paper.

The authors construct a well-designed theoretical background closely related to the current specialised literature in the field.

A short recommendation I would like to made – the theoretical section could be more developed/expanded and the concept of spatial justice could be more explained based on the current recent literature in the field

THE PAPER S STRUCTURE

The structure of the paper is correct in line with the journal standards and meet the publication requirements considering the paper logic. The objectives seem to be clear formulated as well as the investigation is drawn. The core argument of the paper illustrates the paper relevance and the research originality. The results are clearly express and well connected both to the theoretical framework and discussions.

THE METHODS

The methodological design is appropriate and the methods fit well to the present investigation.

I suggest that the all variables and mathematical/statistical determination to be reviewed by an expert in the field of this applied method.

THE MAIN ANALYSIS

The main research is well design and appropriate conducted in line with the main questions in spatial planning in the investigated metropolitan areas. 

CONCLUSIONS

The conclusions fit well summarising the main ideas of the present analysis.

THE GRAPHICAL SUPPORT

The graphical support is well formatted, appropriate illustrating the text content.

THE ENGLISH LANGUAGE

I think the English is ok as far as I could see but I am not a native speaker. I enjoyed to read this paper in English and the language seems well but I think that an opinion of a native English speaker is welcomed. In other words, if the authors used a specialised proofreading services and they could prove this aspect I trust the opinion and the work of this proofread service. On the other hand, I put my trust regarding the English language on the journal editors but I repeat the language seems well.

RECOMMENDATIONS

I am not qualified to assess the mathematical model and determination maybe another reviewer could critically approach it.

I suggest that the explanations from the footer on the pages with the theoretical background to be included in the main text.  

The paper can be published after some minor revision.

Thank you so very much.

Author Response

(The authors gave the same response as above.)

Reviewer 3 Report

This paper investigates the relationship between spatial justice and policy acceptance, which is an interesting topic. I think there are two issues that need to be addressed.

First, I do not like the argument that the offsite realization path is superior to the in-site path. The authors do not from what perspective the offsite realization path is superior, and their justification mainly focuses on the economic side, not focused on spatial justice. Besides, as the authors explained later, whether the offsite path can realize spatial justice depends on whether the compensation is sufficient. Therefore, I believe saying that the offsite path is superior is groundless.

Second, the explanatory variable seems very strange to me as an indicator of spatial justice. How’s that “(Total construction 372 land in 2018- Total construction land in 2013)/Total population in 2013” reflecting the realization degree of spatial justice? This is very confusing. Also, does the realization degree of spatial justice increase or decreases with the value of this indicator, and why? I did not see any explanations. 

Author Response

(The authors gave the same response as above.)

Round 2

Reviewer 1 Report

Some very minor errors in English language appear that should be checked.